# CD44 in Bladder Cancer

**DOI:** 10.3390/cancers16061195

**Published:** 2024-03-18

**Authors:** Jason Duex, Dan Theodorescu

**Affiliations:** 1Samuel Oschin Comprehensive Cancer Institute, Los Angeles, CA 90048, USA; jason.duex@cshs.org; 2Department of Urology, Cedars-Sinai Medical Center, Los Angeles, CA 90048, USA; 3Department of Pathology and Laboratory Medicine, Cedars-Sinai Medical Center, Los Angeles, CA 90048, USA

**Keywords:** CD44, bladder cancer, glycosylation, hyaluronic acid, androgen receptor

## Abstract

**Simple Summary:**

The search to identify and understand proteins which are strongly associated with cancer is critical to our abilities to treat the disease. CD44 is a protein which is known to be strongly associated with cancer progression. While the precise role that CD44 plays in cancer development and resistance to chemotherapy agents remains to be fully determined, the protein provides three avenues for possible novel therapeutic approaches. First, since CD44 is found on the surface of cancer cells, it is more accessible to the therapeutic agents which block its ability to carry out its negative functions. Second, since CD44 binds a sugar moiety called hyaluronic acid, chemotherapy agents can be packaged with hyaluronic acid so that they preferentially bind cancer cells with high levels of CD44 on their surface, resulting in a greater efficacy in treatment. Third, the levels of CD44 can be assessed in urine voided from bladder cancer patients, possibly informing on the aggressiveness or stage of the cancer without the need for invasive testing. All told, CD44 protein provides a number of opportunities to better monitor and therapeutically target cancer and deserves increased resources to further clarify this potential.

**Abstract:**

The glycoprotein CD44, with its many isoforms and variations in carbohydrate patterning, participates in a diverse set of cellular functions. This fact leads to the protein playing a role in many normal and pathologic cellular processes including a role in cancer progression and metastasis. These same facts make CD44 a strong therapeutic target in many cancer types, including bladder cancer.

## 1. Introduction

CD44 is a cell-surface protein expressed in most mammalian cell types [1]. The protein is found in different forms and is involved in a wide array of normal cellular functions such as blood cell development, lymphocyte activation, and homing. It is also found in a wide array of abnormal biological functions within the human body, such as tumor growth and metastasis [2]. In normal biology, CD44 has been shown to regulate cellular metabolism, in particular glucose metabolism [3,4,5] and angiogenesis [6]. These normal biological roles for CD44 also make its overexpression highly beneficial when it occurs in cancer cells [7]. The ability of one protein to be involved in so many different cellular processes is the result of the many different forms of CD44 that have been found to exist in cells [8].

The role of the bladder is to store urine. Environmental carcinogens from cigarettes, for example, cause most bladder cancers [9]. Bladder cancer presents in two forms. The most common is non-muscle-invasive bladder cancer, which remains in the luminal epithelial (urothelial) cell layer and has not yet penetrated the muscle layer of the bladder [10]. Non-muscle-invasive bladder cancer is treated by means such as transurethral resection, with and without clinical instillation of therapeutics [9]. Approximately one third of patients present with muscle-invasive bladder cancer, which penetrates the muscle cells surrounding the bladder [10]. This advanced form has a worse prognosis and requires major surgery or radiation for treatment [9]. This review will discuss the relationship between CD44 and bladder cancer biology and patient outcome and its utility as a diagnostic, prognostic, and predictive marker.

## 2. CD44 Isoforms

The CD44 gene is composed of 20 exons, located on chromosome 11. Exons 1–5 encode the extracellular domain, while exons 16–20 encode the transmembrane domain and the intracellular domain. Three recent reviews provide excellent illustrations of the CD44 domains and the numerous spliced variants that are possible from the CD44 gene [7,11,12]. Exons 6–14 encode for a variable region in the protein, located in the extracellular domain. Complex alternative splicing of exons 6–15 then leads to a wide variety of CD44 isoforms, and the NCBI database lists eight CD44 isoforms that have been detected biologically in humans. The standard CD44 isoform lacks exons 6–15 of the variable region. Thus, when an exon, or exons, of the variable region is spliced into the CD44 messenger RNA, it results in a protein with additional residues in the extracellular domain.

The isoforms are named according to the variable domain exons (exons 6–15) that are present in the isoform, with an isoform which has the first variable domain exon (exon 6) spliced in being called variant 1. The most commonly identified isoforms of CD44 to date are CD44v3, CD44v6, CD44v9, and CD44v8–10 [13,14]. These isoforms are a significant source of CD44 diversity found in human cells, and changes in the populations of these isoforms can have a profound effect on cell growth. In fact, it is in cancer cells that the alternatively spliced CD44 protein isoforms have been found to be expressed at their highest levels [7].

## 3. Glycosylation and CD44

Another driver of CD44 diversity is the array of glycan molecules which decorate its extracellular domain [15]. Glycan molecules are composed of polymerized monosaccharides, that are in turn linked via covalent bonding to another molecule and can vary in composition, branching, and size [16]. In humans, there are 17 different types of monosaccharide units that are commonly found conjugated to specific proteins [17]. The attachment of these glycan molecules to proteins and lipids is called glycosylation and occurs in the endoplasmic reticulum after the protein has been translated. The glycosylation event is in fact the most common form of posttranslational modification that a protein undergoes in humans [18]. Glycosylation is a process so critical to cellular function that there exist nearly 700 genes in the human genome that are associated with this process [18]. This means that roughly 3% of the human genome is dedicated to managing this process, illustrating its importance in the various functions of a human cell.

There are 12 major types of glycosylation that occur in humans [17]. Two of the most common types are N- and O-glycosylation. N-glycosylation occurs when glycan molecules are attached to the amino acid asparagine (Asn or N). Another linkage occurs on the hydroxyl group of serine or threonine amino acids and is thusly termed O-glycosylation. The glycosylation of proteins can serve both general and specific roles. General roles include helping certain proteins fold correctly into their tertiary structure or helping to stabilize certain proteins. Another general role of glycosylation of cell-surface proteins and lipids is to form a protective barrier for the cell from stressful events in its environment, as well as helping cells maintain shape [19,20,21]. Specific roles for glycosylation include modulating the degree to which cells adhere to matrices, regulating cell-surface receptor dimerization, effector function within the innate immune system, and ligand binding [18]. In fact, variations in the degree of N-glycosylation that occur on CD44 can enhance or block the ability of that protein to bind certain ligands. Most notably, the addition of exons during the splicing phase of transcription from the CD44 gene can lead to additional asparagine and serine or threonine amino acid residues in the mature CD44 protein. These residues can provide additional locations on CD44 for N- and O-glycosylation. As such, these additional glycosylation additions can lead to the inability of these atypical CD44 proteins to bind typical effector molecules or promote their ability to perform atypical functions [22].

## 4. Glycosylation and Bladder Cancer

Glycosylation is critical for optimal functioning of the human immune system, not only for proper response of the immune system in the host, but also as a pattern to be recognized on foreign species. Alterations in the glycosylation of proteins outside the immune system can lead to those proteins becoming pro-transforming agents. The most well documented example to date is the MUC1 protein, which normally functions, when properly glycosylated, as a coactivator of transcriptional elements related to metabolic functions [23]. MUC1 overexpression has been found in many types of cancer [23], and is certainly found in bladder cancer [24]. Expression analysis has revealed that MUC1 expression is significantly increased in bladder tumor tissues, as well as lymphatic metastases [25]. More concerning from a therapeutic side is that MUC1 proteins were found to play a significant role in bladder cancer cells acquiring resistance to cisplatin, one of the major chemotherapeutic agents used to treat bladder cancer patients [26]. Additional studies have found that the degree of glycosylation of MUC1 may be critical to its role in cancer. For example, MUC1 under glycosylation is associated with more advanced disease in many types of cancer [27,28]. In other cases, altered expression systems in cancer cells lead to an incompleteness in normal MUC1 glycosylation, leading to the formation of atypical side chains and an MUC1 protein which promotes disease [29].

The role of general glycosylation levels in the cellular proteome of bladder cancer cells specifically is becoming clearer with recent studies. It is evident that alterations to a number of glycan synthesis pathways are associated with bladder cancer occurrence, more advanced disease, and poor prognosis [30]. Another hallmark of abnormal glycan synthesis and protein modification in bladder cancer is the increased appearance of Tumor-Associated Carbohydrate Antigens (TACAs) [31,32]. For example, the Tn antigen is a TACA that is not found in normal tissues, but due to decreases in the normal O-glycosylation pathway this antigen is created [33]. Additional TACAs occur as a result of different saccharide molecules being added in an O-glycosylation-deficient environment, such as STn, T, and Lewis antigens. All of these abnormal antigens are considered significant biomarkers in bladder cancer, along with a host of other altered glycosylation events [30]. A very intriguing, recent development in assisting clinicians with choosing the best treatment for bladder cancer patients is the development of a glycosylation risk score [34]. Among the more intriguing elements of this approach is its potential for identifying the efficiency of immune cell infiltration and efficacy of immunotherapies such as immune checkpoint blockade.

## 5. CD44 Ligand Binding and Effectors

CD44 is also a cell-surface receptor which binds an array of ligands, including hyaluronic acid (HA), collagens, osteopontin (OPN), matrix metalloproteinases, Siglec-15, fibronectin, TM4SF5, PRG4, FGF2, and podoplanin, and activates or inhibits c-Src/STAT3/Twist1/Bmi1, PI3K/AKT/mTOR, ERK/NF-κB/NANOG and other signaling pathways [5,15]. Owing to these various interactions, and its presence on the surface of cells, CD44 plays a role in cell growth, cell–cell interactions, cellular adhesion, and cellular migration in healthy and disease tissue. When CD44 is activated, the intracellular domain translocates to the nucleus to promote transcriptional changes associated with cell cycle and metabolism programs via the regulation of c-Src/STAT3/Twist1/Bmi1, PI3K/AKT/mTOR, ERK/NF-κB/NANOG, and other signaling pathways [15,35,36].

One of the main ligands for CD44 is HA. It is worth noting that as glycosylation is the addition of saccharide molecules to proteins and lipids, HA is itself a free oligosaccharide, unbound to any protein or lipid. As discussed above, glycosylation can block the ability of a protein to bind ligands. A relevant example is that N-glycosylation levels on CD44 control the affinity of CD44 for the ligand HA [22].

## 6. CD44 as a Marker for Bladder Cancer Growth and Progression

One of the most common uses of CD44 in the cancer field is with CD24 as cell-surface markers of breast [37] and other cancer stem cells [38]. In bladder cancer CD44 expression on the surface of cells is associated with a more aggressive cellular phenotype and has been used to stratify patient prognosis [39,40,41,42,43,44] (Table 1). Another use of CD44 as a marker is to define cellular subtypes in bladder cancer [9], which has been shown to stratify patient prognosis and therapeutic response in muscle-invasive disease [9,10]. Luminal subtype bladder cancer is defined by cellular markers Uroplakin II and CK20, while basal bladder cancer cells are defined most by CK5/6 and CD44 [45,46,47].

There is a strong association between CD44 expression and bladder cancer clinical stage, lymph node metastasis [42], as well as disease-specific survival [40]. One likely role for CD44 as a driver of tumor growth is as a cell receptor responding to extracellular cues relayed via the various ligands which bind CD44. CD44 is a major receptor for hyaluronic acid (HA), and the partnership of these two molecules is strongly associated with cancer [53,54,55,56] and viral infections [57]. The relationship between HA and bladder cancer is also strong. First, higher HA levels are associated with bladder cancer [58,59]. Second, the HA synthase gene HAS2 was found to have elevated expression in bladder cancer cells [59,60,61]. Third, expression levels of HYLA1 (hyaluronidase 1), an enzyme which regulates HA levels, are associated with bladder cancer recurrence and advancement to the muscle-invasive stage [61,62]. Lastly, the small molecule 4-Methylumbelliferone, which blocks synthesis of HA, inhibits bladder cancer cell growth in vitro and in vivo [59]. Taken together, blocking production of HA, one of the main ligands for CD44, inhibits growth of bladder cancer cells in vitro and in mouse xenograft models, supporting the notion that CD44 functioning as a receptor is a driver of bladder cancer growth, progression, and recurrence.

In addition to detecting the presence of CD44 on the cell surface and quantifying the levels of the protein, additional studies are looking at transcriptomics to identify and eventually quantify the presence and levels of CD44 isoforms being expressed in bladder cancer cells [50]. In an analysis of clinical bladder cancer samples, the presence of CD44v9 showed a particularly strong correlation with worse prognoses [44]. Additionally, expression levels of CD44v6 and CD44v3 were found to be twice as high in tumor tissue relative to adjacent normal tissue in 50 bladder cancer patient samples [49]. This same study also demonstrated, using in vitro studies, that silencing of the CD44v3 isoform leads to reduced activation of some common signaling molecules and reduced cell growth, hinting at a potential direct role in cancer cell growth for CD44 isoforms. However, it should be noted that correlation of CD44 isoforms with bladder cancer clinical outcome associations requires further validation [13].

Studies have also shown that overexpression of CD44 promotes disease progression by enhancing chemoresistance and anti-apoptotic programs in cancer cells [63]. For example, high expression levels of CD44 are associated with increased expression of chemoresistance proteins, including the classic chemoresistant protein Multidrug Resistance Protein 1 (MDR1) [64,65]. Additionally, a high expression of CD44 promotes expression of the anti-apoptotic programs associated with MCL-1 and Survivin [66], Nanog [67], and the PI3K/AKT pathway [68]. In bladder cancer, a direct role for CD44 in promoting chemoresistance has not been discovered. However, bladder cancer stem cells are highly resistant to chemotherapeutic drugs, and the expression of CD44 standard length and the CD44v6 isoform are commonly associated with bladder cancer stem cells [69].

The identification of osteopontin (OPN) as a ligand for CD44 was made in 1997 [70] and has been implicated in a number of physiological and pathological conditions including metastasis of nasopharyngeal carcinoma [71] and bladder cancer [52], as well as control of CD8+ T cell activation and tumor immune evasion [72]. The connection to bladder cancer was made though the observation that RhoGDI2, a metastasis suppressor gene of relevance in bladder cancer [73], specifically suppresses metastasis by reducing tumor-associated macrophages in the tumor microenvironment [74]. In this study, macrophage-secreted osteopontin was found to bind to a short version of CD44 (CD44s) on the tumor cells and promote bladder cancer invasion and growth. These effects were RhoGDI2-sensitive and required CD44s binding to the Rac GEF TIAM1 signaling molecules. Further investigation of the CD44 ligands found osteopontin to be the driver of this process, which was supported by finding its expression correlated with tumor aggressiveness and a poor clinical outcome in bladder cancer patients. Inhibiting this pathway potently blocked lung and lymph node metastasis in animal models while primary tumors and established metastasis were found to be less sensitive. These findings showed that the osteopontin–CD44–TIAM1–Rac1 axis is a RhoGDI2-sensitive pathway and potential therapeutic target in bladder cancer metastasis.

CD44 has been found to have a relationship with the male hormone androgen and its receptor in bladder cancer. The androgen receptor (AR) is important in the development of both experimental and human bladder cancer, yet the role of AR in bladder cancer growth and progression is less clear, with the literature indicating that more advanced stage and grade disease are associated with reduced AR expression [75,76]. Recent chromatin immunoprecipitation sequencing and transcriptomic approaches investigating AR discovered that CD44 was significantly associated with androgen effects [51]. CRISPR-based mutagenesis of the AR response elements associated with CD44 identified a novel silencer element leading to the direct transcriptional repression of CD44 expression. In human patients with bladder cancer, tumor AR, CD44 mRNA, and protein expression were inversely correlated, suggesting a clinically relevant AR–CD44 axis. This work suggests a novel mechanism partly explaining the inverse relationship between AR and bladder cancer tumor progression and suggests that AR and CD44 expression may be useful for prognostication and therapeutic selection in primary bladder cancer.

## 7. CD44-Based Detection and Monitoring of Bladder Cancer

As discussed above, the degree of glycosylation in bladder cancer tissues is an emerging marker for assessing bladder cancer disease. Additionally, there are the TACAs which are found in bladder cancer tissues but not normal tissues, as well as glycan heavy molecules such as MUC1 whose levels correlate strongly with bladder cancer presence and progression. Detecting these events for a disease of the bladder has a distinct advantage. Bladder cancer is a result of the transformation of the luminal urothelial cells that line the bladder cavity, hence analyzing the voided urine of patients for cancer markers is an excellent, unique, and non-invasive means of monitoring progression in bladder cancer patients. As discussed above, CD44 has been identified as a protein which is strongly correlated with disease state in bladder cancer. One developed test used a glycan affinity platform to isolate glycan-containing molecules from the urine of control and bladder cancer patients. Proteins found to be enriched in bladder cancer patients relative to control patients, were investigated for their correlation with grade of disease [77]. CD44 was found to be significantly enriched in bladder cancer patients. Orthogonal studies demonstrated that CD44 protein levels in the urine were significantly and directly correlated with high-grade and muscle-invasive bladder cancer disease. Another study isolated N-glycopeptides from the urine of non-muscle-invasive bladder cancer and muscle-invasive bladder cancer patients. These analyses identified unique glycosylation patterns of CD44 in muscle-invasive patients compared to non-muscle-invasive patients [78]. Such a finding points to the possibility of frequent, non-invasive screening of patients with manageable, non-muscle-invasive bladder cancer to monitor for progression of the disease to a muscle-invasive state, which would require a more aggressive treatment program. Progression-free survival of bladder cancer patients was also shown to correlate with expression levels of CD44 in patient urine [79]. In fact, CD44 was found to be one of the main differentially expressed genes between bladder cancer patients and control urine samples. The study’s authors identified CD44 among a six gene panel, which had high prognostic value relating to the overall survival of bladder cancer patients. Another study developed a means to quantify expression levels of the CD44 variant CD44v6 in cells found in urine. This study found that in cells voided in the urine there was a statistically significant increase in CD44v6 expression in bladder cancer patients, relative to control individuals [80]. All told, multiple studies have found CD44 to correlate with bladder cancer presence when examining this in urine, highlighting the importance of this protein, and its expression levels, in diagnostic methods for bladder cancer.

## 8. CD44 Pathway-Based Therapy

There are several therapeutics being developed that take advantage of CD44 protein levels commonly being higher in bladder cancer compared to normal urothelium and that CD44 is a receptor for HA [81] (Table 2). A promising approach involves the attachment of HA molecules to gold (Au)-augmented silicon dioxide (SiO_2_) particles. This results in a dramatic increase in delivery of the HA-AuSiO_2_ particles only to bladder cells expressing CD44. Subsequent photothermal therapy of cells exposed to the HA-AuSiO_2_ treatment involves heating the AuSiO_2_ particles to induce the death [82] of cells which acquired the heavy metal therapeutic. Another approach targets HA production in bladder cancer patients. The HA synthase inhibitor, 4-methylumbelliferone (4-MU), would reduce production of HA, thereby reducing activation of CD44. Preclinical studies have shown that a subset of cancer cells, which have lost expression of the glycogen debranching enzyme (AGL) that cells use in metabolism, are highly sensitive to treatment with 4-MU [83]. In fact, 4-MU has been shown to be effective against several cancer types [84] and is well tolerated by patients in a clinical trial (NCT00225537).

Targeting the CD44 protein itself is yet another approach used in the pre-clinical space. Pre-clinical studies have shown that CD44 knockdown diminishes the ability of cancer cells to enter glycolysis and enhanced the efficacy of chemotherapeutic drugs [4]. One way to target CD44 directly is to use synthetic peptides or single stranded oligonucleotides called aptamers, with in vitro and mouse in vivo studies showing significant efficacy [90]. CD44 can also be directly targeted with a monoclonal antibody. In fact, humanized monoclonal antibodies which target CD44 have been developed and have been shown to reduce bladder cancer tumors in mouse models [89].

HA-based therapeutics have been developed for clinical applications. HA-dexamethasone nanoparticles have been shown to strengthen the delivery of the anti-inflammatory dexamethasone to site-specific locations in patients suffering from a variety of conditions, including hearing loss [91,92], osteoarthritis of the knee [93], lung inflammation, and other disorders [94]. For bladder cancer, HA-based clinical applications include conjugating chemotherapeutic agents to HA, thereby allowing cells with high levels of HA receptors, such as CD44, to uptake the agents. This approach spares healthy cells from toxic agents. HA-paclitaxel has been developed (commercial name ONCOFID-P-B) and is being investigated in bladder cancer patients in a Phase III clinical trial (NCT05024773). HA-doxorubicin and HA-cisplatin are other examples of successful conjugates that are being tested for their anti-tumor effect but are still in the early stages of investigation [95].

Non-muscle-invasive bladder cancer, which accounts for approximately 70%, is generally not fatal, and patients are usually treated by local means without removal of the bladder [9]. Approaches include local resection or administering therapeutics directly into the lumen of the bladder with a catheter. Given the biologically relevant physiochemical properties of HA polymers to mimic the extracellular matrix and form organic carrier molecules (e.g., liposomes and nanoparticles), they can be engineered into hydrogels and enriched with specific cargos [81]. Additionally, given the sticky nature of hydrogels, they also increase the residency time of any treatment when they are administered to the bladder lumen at the same time as a chemotherapeutic agent in intravesical delivery. Thus, another HA-based approach is to package the agents in hydrogels, and such approaches are currently being investigated for their ability to deliver anti-bladder cancer drugs such as iron oxide nanoparticles [87] and various chemotherapeutic agents [96]. Such an approach provides two major benefits to the bladder cancer patient. First, the dwell time of agents in the bladder and against the wall of the bladder is increased due to increased viscosity of the hydrogel matrix. Second, delivery of the agents inside the bladder cancer cells in the patient has been demonstrated to be increased with the use of hydrogels due to cell adhesion and the increased cellular permeability owing to the amphipathic properties of the hydrogels [97,98]. Chemotherapeutic agents have low specificity for tumor tissues and low solubility in the blood stream. These negative aspects result in high dosing to achieve efficacy and harmful side effects for the patients [99]. The use of HA-based therapeutic delivery mechanisms will be important in taking chemotherapeutics to the next level for patients.

As discussed above, targeting the CD44 protein itself has shown promise to counter the growth of bladder cancer cells in vitro and in mouse models. A humanized anti-CD44 antibody has in fact been tested on patients in the clinic. This anti-CD44 antibody was tested in a Phase I clinical trial (NCT01358903) in patients with CD44-expressing tumors from a variety of histologies [100]. The results from this study did not warrant a follow-on study. However, it was performed on patients with metastatic and locally advanced, solid tumor disease. Future anti-CD44 investigations should look at earlier bladder cancer stages and in non-muscle-invasive disease, with the advantage of the latter being the ability to deliver the therapy intravesically, which will likely reduce side effects associated with systemic monoclonal antibody therapy.

## 9. Conclusions and Future Directions

Numerous studies over the past two decades have shown that CD44 is important in cancer cell identification, behavior, and as a prognostic factor in patients. However, this importance has not been actualized into clinically effective therapies despite promising in vitro and preclinical data. As discussed above, the CD44 protein itself can be targeted with several agents, from antibodies to synthetic peptides to aptamers. Targeting HA production could also lead to reduced CD44 activation. Additionally, treatments can also employ HA-based therapies, such as HA-containing nanoparticles or HA-conjugated anti-cancer agents. If single treatments were not adequately effective, combining several that would attack various aspects of the CD44 signaling pathway may be more effective. If any technical or drug interactions arise in using the agents simultaneously, alternating them in cycles may avoid these issues while still improving efficacy. There are several challenges associated with systemic administration of these therapies, as these components (e.g., antibodies or engineered cells like in CAR-T therapies) may have nonspecific effects or other toxicities. Because of its location, bladder cancer has the advantage of intravesical administration of such treatments in a recurring fashion, which offers a potential means of targeting bladder cancer while also limiting the potential for related adverse events.

One significant area to focus on in future patient analyses and laboratory experiments is the association between CD44 and immunotherapies. Recent therapeutic advances, particularly the immune checkpoint inhibitor (ICI) line of therapies, have resulted in favorable response rates in many patients with cancer. Bladder cancer patients are responding significantly to these therapies as well [101]. A recent study investigated expression of CD44 and potential relationships with elements of 33 different cancer types that make them more susceptible to ICI treatments, including tumor mutational burden (TMB) and microsatellite instability. CD44 exhibited strong correlations with indicators of survival and positive prognoses [102]. In bladder cancer, CD44 was associated with high TMB. Furthermore, expression data from the Tumor Immune Dysfunction and Exclusion (TIDE) database, which includes expression data from bladder cancer patients treated with the ICI therapeutic anti-PD-L1, revealed that TMB was associated with the best response among nearly all other cancers. This study also found that CD44 expression in bladder cancer patient samples was significantly associated with tumor infiltration by immune cells. Another study, using similar analyses and patient datasets, came to the same conclusion [48]. This latter study noted the association between the PD-L1 protein and CD44 in the patient samples and observed in laboratory experiments that knockdown of CD44 led to a decrease in PD-L1 expression.

While the existing data is supportive of CD44 playing a significant role in cancer progression and response rates to ICIs, including bladder cancers, the mechanistic details behind this role remain limited. Hence, more mechanistic studies would provide much needed insight into the degree to which CD44 itself is potentially driving aggressive and therapy-resistant disease.

## Figures and Tables

**Table 1 cancers-16-01195-t001:** CD44 and Bladder Cancer.

Involvements	Findings	Methods	References
Defines bladder cancer cell subtype	Basal bladder cancer cell type	IHC mRNA (NanoString)	[45,46,47]
Regulates immune cell infiltration	Gene expression correlation	Transcriptomics IHC	[48]
Splice variants CD44 v3, v6, v9	Invasion and poor prognosis	Transcriptomics Glycoproteogenomics IHC	[44,49,50]
Correlates with lymph node metastasis	Higher expression and protein levels correlate with nodal invasion	Transcriptomics IHC	[42,43]
Disease specific survival	Higher expression associated with decreased bladder cancer patient survival	Transcriptomics IHC IHC IHC	[39,40,43]
Disease recurrence	Higher protein levels associated with disease recurrence	IHC	[41]
Progression to invasive disease	Higher protein levels correlate with deep muscle infiltration	IHC	[43]
Association with AR	Correlates with disease	Transcriptomics	[51]
Binding of osteopontin	Correlates with tumor aggressiveness and poor clinical outcome	Transcriptomics	[52]

**Table 2 cancers-16-01195-t002:** CD44 Pathway Directed Therapies in Bladder Cancer.

Approach	Target	Study Types	Reference
HA-based delivery of siRNA molecules	CD44	In vitro Mouse model	[85]
HA-AuSiO_2_	HA	In vitro Mouse model	[82]
HA synthase inhibitor 4-MU	HA	In vitro Mouse models Patients	[83,84] NCT00225537
Aptamer competitive inhibitors	CD44	In vitro Mouse models	[86]
HA-based hydrogels–iron oxide nanoparticles	Intravesical dwell time increase	Mouse model	[87]
HA-based hydrogels–cisplatin conjugation	Intravesical dwell time increase	Rat model	[88]
Neutralizing antibody	CD44	In vitro Mouse models Patients	[89] NCT01358903
HA-paclitaxel	HA	Patients	NCT05024773

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
