# Peer review of "CD44 in Bladder Cancer"

_cancers, 2024, doi:10.3390/cancers16061195_

Round 1

Reviewer 1 Report

Comments and Suggestions for Authors

The abstract should include the objective and the study design.
The concept of "abnormal biology" is very imprecise. The authors must be clear when they are referring to cell biology. If they want to refer to human individuals, then mentioning physiological conditions or pathophysiology is preferable.
A very extensive section talks about glycosylation regarding CD44 and also Bladder cáncer in general.
The review's aim is expressed in the last three rows of the introduction section and includes discussing the utility of CD44 as a diagnostic, prognostic, and predictive marker. However, the authors fail to give information that includes values such as specificity, sensitivity, predictive values, and likeñihoo0d ratios to approach correctly. Instead, they are giving very general information that could be an overview of its potential clinical role.
The manuscript should be restructured regarding those points.

Author Response

  • The abstract should include the objective and the study design.

In the instructions to authors there are no instructions that align with this reviewer suggestion.

Cancers | Instructions for Authors (mdpi.com)

Also, in looking at the first 5 reviews in the latest issue, none of them have the objective or study design. Nearly all published reviews in this journal have an abstract that lacks any sections or section labels.

Cancers | An Open Access Journal from MDPI

  • The concept of "abnormal biology" is very imprecise. The authors must be clear when they are referring to cell biology. If they want to refer to human individuals, then mentioning physiological conditions or pathophysiology is preferable.

This is a good point. The word “biology” has been replaced with “biological functions within the human body”. The sentence now reads, “It is also found in a wide array of abnormal biological functions within the human body, such as tumor growth and metastasis”.

  • A very extensive section talks about glycosylation regarding CD44 and also Bladder cáncer in general.

It is not clear to us if this is a criticism of the section, but another reviewer commented on the length of this section, relative to the topic of CD44 and bladder cancer. We have removed two large paragraphs (~265 words) from this section and left only the sections which discuss glycosylation and bladder cancer directly.

  • …the authors fail to give information that includes values such as specificity, sensitivity, predictive values, and likeñihoo0d ratios to approach correctly. Instead, they are giving very general information that could be an overview of its potential clinical role.

We feel this is beyond the scope of this review.

Reviewer 2 Report

Comments and Suggestions for Authors

A Beautiful review! 

1. It would be perfect if the author can add a figure to show the difference of CD44 isoforms.

2. The author clarifies different reports of CD44 isoforms in bladder cancer, such as CD44v6 in bladder cancer stem cells [1], CD44v9 on bladder cancer bladder prognosis [2]. It would be beautiful if the author can categorize different role of these isoforms in bladder cancer as a form.

Reference:

1. Li Y., Lin K., Yang Z., Han N., Quan X., Guo X., Li C. Bladder cancer stem cells: clonal origin and therapeutic perspectives. Oncotarget. 2017; 8: 66668-66679.

2. Kobayashi K, Matsumoto H, Matsuyama H, Fujii N, Inoue R, Yamamoto Y, Nagao K. Clinical significance of CD44 variant 9 expression as a prognostic indicator in bladder cancer. Oncol Rep. 2016 Nov;36(5):2852-2860.

Author Response

  • A Beautiful review!

Thank you very much.

  • It would be perfect if the author can add a figure to show the difference of CD44 isoforms.

Given the brevity of the review, and that it focuses on bladder cancer as much as CD44, if not more, we feel the isoform figure is not necessary. There are 3 very recent reviews on CD44 isoforms which include excellent illustrations (from 2021, 2022, and 2023). These reviews are referenced in our CD44 and bladder cancer review.

  1. CD44: A Multifunctional Mediator of Cancer Progression 2021
  2. Role of CD44 isoforms in epithelial-mesenchymal plasticity and metastasis 2022
  3. RNA-binding proteins regulating the CD44 alternative splicing 2023

However, we do feel that highlighting the figures would be beneficial to the readers, so we have now added the following sentence to the review. Thank you for the feedback:

“Three recent reviews provide excellent illustrations of the CD44 domains and the numerous spliced variants that are possible from the CD44 gene [PMID: 34944493, PMID: 35023031, PMID: 38106992].”

  • The author clarifies different reports of CD44 isoforms in bladder cancer, such as CD44v6 in bladder cancer stem cells [1], CD44v9 on bladder cancer bladder prognosis [2]. It would be beautiful if the author can categorize different role of these isoforms in bladder cancer as a form.
  • Reference:
    • Li Y., Lin K., Yang Z., Han N., Quan X., Guo X., Li C. Bladder cancer stem cells: clonal origin and therapeutic perspectives. Oncotarget. 2017; 8: 66668-66679.
    • Kobayashi K, Matsumoto H, Matsuyama H, Fujii N, Inoue R, Yamamoto Y, Nagao K. Clinical significance of CD44 variant 9 expression as a prognostic indicator in bladder cancer. Oncol Rep. 2016 Nov;36(5):2852-2860.

We understand the reviewer suggestion, and think it is a good one. However, there is only one published study investigating the potential role(s) of CD44 variants in bladder cancer. This potential role was investigated in vitro, and we’ve tried to keep the review focused on clinical samples as much as possible. Additionally, this lone study used just one cell line. This is not to criticize the study, but we were hesitant to include such limited experimental info in this short review. With that said, we did add a sentence regarding the potential role of the CD44v3 from that study, which will also hopefully relay to the reader that there is not a lot known about the role(s) of CD44 variants in bladder cancer cells at this time.

“This same study also demonstrated, using in vitro studies, that silencing of the CD44v3 isoform leads to reduced activation of some common signaling molecules and reduced cell growth, hinting at a potential direct role in cancer cell growth for CD44 isoforms.”

Interestingly, CD44v6 silencing had no effect.

Reviewer 3 Report

Comments and Suggestions for Authors

The manuscript by Duex et.al. provided a comprehensive and updated summary of the role CD 44 in the biology of bladder cancer as well as CD 44 targeted therapies. It is well organized and clearly written in a way that is easy for readers to follow. My only comment is it would be nice for authors to comment on the implication of CD 44 in the era of immunotherapy for bladder cancer. One example is PMID: 37316699 and I am sure there are multiple similar studies in the literature. This will make the current manuscript more clinically relevant considering the current paradigm shift in the management of bladder cancer. 

Author Response

  • The manuscript by Duex et.al. provided a comprehensive and updated summary of the role CD 44 in the biology of bladder cancer as well as CD 44 targeted therapies. It is well organized and clearly written in a way that is easy for readers to follow. My only comment is it would be nice for authors to comment on the implication of CD 44 in the era of immunotherapy for bladder cancer. One example is PMID: 37316699 and I am sure there are multiple similar studies in the literature. This will make the current manuscript more clinically relevant considering the current paradigm shift in the management of bladder cancer.

This is a good suggestion. Unfortunately, there is not a lot of literature yet on association of CD44, bladder cancer, and immunotherapies. However, we did add 260 words to the Conclusions and Future Directions section of the review to discuss what little is known.

Reviewer 4 Report

Comments and Suggestions for Authors

This is a generally well written and comprehensive review of the role of CD44 in bladder cancer.  The review fills a need for a review pulling together the clinically and scientific topic of CD44 and bladder cancer. The topics reviewed are mostly relevant. However, the extensive general discussion of glycosylation (Lines 90 - 143) seems irrelevant to the main topic and too elementary for this review.

The review repeats a flaw for readers that is all too common in review articles, namely the presentation of findings in qualitative terms such as "expression levels were correlated with disease severity." Such reporting would be ever so much more useful, particularly for discussing biomarkers, if the results were presented in more quantitative terms such as sensitivity and specificity, correlation coefficients and study size.  

I noted a few minor typos as noted below.

Line 32 TUR is not noninvasive.

Line 45 "does is" is redundant

Line 202 add ly to common

Line 240 Capitalize bladder

Author Response

  • This is a generally well written and comprehensive review of the role of CD44 in bladder cancer. The review fills a need for a review pulling together the clinically and scientific topic of CD44 and bladder cancer. The topics reviewed are mostly relevant. However, the extensive general discussion of glycosylation (Lines 90 - 143) seems irrelevant to the main topic and too elementary for this review.

Thank you for the feedback. Another reviewer also commented on the length of this section, relative to the topic of CD44 and bladder cancer. As such, we have removed two large paragraphs (~265 words) from this section and left only the sections which discuss glycosylation and bladder cancer directly.

  • The review repeats a flaw for readers that is all too common in review articles, namely the presentation of findings in qualitative terms such as "expression levels were correlated with disease severity." Such reporting would be ever so much more useful, particularly for discussing biomarkers, if the results were presented in more quantitative terms such as sensitivity and specificity, correlation coefficients and study size.

We feel this is beyond the scope of this review.

  • I noted a few minor typos as noted below.
    • Line 32 TUR is not noninvasive.

This is a good point. The phrase non-invasive has been removed from the sentence.

  • Line 45 "does is" is redundant

Now corrected

  • Line 202 add ly to common

Now corrected

  • Line 240 Capitalize bladder

Now corrected

Round 2

Reviewer 1 Report

Comments and Suggestions for Authors

The authors addressed correctly mentioned issues